# The Machine Learning Methods in Non-Destructive Testing of Dynamic Properties of Vacuum Insulated Glazing Type Composite Panels

**DOI:** 10.3390/ma16145055

**Published:** 2023-07-17

**Authors:** Damian Kozanecki, Izabela Kowalczyk, Sylwia Krasoń, Martyna Rabenda, Łukasz Domagalski, Artur Wirowski

**Affiliations:** 1Department of Structural Mechanics, Lodz University of Technology, Politechniki 6, 93-590 Lodz, Poland; damian.kozanecki@dokt.p.lodz.pl (D.K.); izabela.kowalczyk@dokt.p.lodz.pl (I.K.); sylwiakrasonn@gmail.com (S.K.); lukasz.domagalski@p.lodz.pl (Ł.D.); artur.wirowski@p.lodz.pl (A.W.); 2Department of Concrete Structures, Lodz University of Technology, Politechniki 6, 93-590 Lodz, Poland

**Keywords:** vacuum glazing, dynamic analysis, artificial intelligence, neural networks, extreme gradient boosting, machine learning

## Abstract

The VIG (Vacuum Insulated Glazing) unit, composite glazing in which the space between glass panes is filled with vacuum, is one of the most advanced technologies. The key elements of the construction of VIG plates are the support pillars. Therefore, an important issue is the analysis of their mechanical properties, such as Young’s modulus and their variability over a long period of time. Machine learning (ML) methods are undergoing tremendous development these days. Among the many different techniques included in AI, neural networks (NN) and extreme gradient boosting (XGB) algorithms deserve special attention. In this study, to train selected methods of machine learning, numerical data developed in the VIG plate modelling process using Abaqus program were used. The test method proposed in this article is based on the VIG plate subjected to forced vibrations of specific frequencies and then the reading of the dynamic response of the composite plate. Such collected and pre-developed experimental data were used to obtain the mechanical parameters of the steel elements located inside the analysed vacuum glazing. In the future, the proposed research methods can be used to analyse the mechanical properties of other types of composite panels.

## 1. Introduction

### 1.1. Purpose and Scope of Work

This is the first paper to describe the Machine Learning (ML) models that are capable of predicting mechanical parameters of VIG-type plates based on their mechanical and dynamic properties.

The work could be divided into four steps. The first step was to create a numerical imitation of the VIG plates using FEM software (Abaqus CAE 2017) and to obtain natural frequencies for assumed sets of numerical models with different geometrical and mechanical parameters. The second step was to train and test the predictive model using data collected from the numerical analysis. The third step was to carry out experimental modal analysis for seven different VIG plates in order to obtain their first thirty natural frequencies.

The final step was to use a predictive ML model to anticipate mechanical properties of support pillars located inside experimentally tested plates.

### 1.2. Motivation

There are many studies concerning VIG-type windows, most of them focused on aspects referred to as thermal and acoustic insulation, as well as mechanical properties. In [1], Simko investigated the heat transfer processes and stresses in VIG panels. Cho and Kim tested the heat transfer coefficient of vacuum glazing using a simulation in which they considered multiple cases that included different glass types, number of layers and spacing between pillars [2]. Arya and Hyde theoretically analysed a vacuum glazing system with rigid and flexible edge seals [3], while Ashmore, Cabrera, and Kocer compared the acoustic performance of VIG and monolithic glazing [4]. Nonetheless, in [5], the authors analysed the effect of different geometric and mechanical parameters on the static and dynamic response of VIG panels.

As every part of the building, vacuum glazing windows are exposed to external factors, such as rainfall or sunbeams or wind forces. As the result, they degrade over time and there might be a need to check their current condition. The verification of the vacuum glazing components’ mechanical properties located in the existing buildings is impossible. This is due to the fact that the structure of the window cannot be damaged during this process. Therefore, a non-destructive method is required.

### 1.3. VIG Panels Description

The 21st century brings many challenges that provide changes in mindset of civil engineers. Progressive climate change has become a motivation to modify design guidelines. Due to the need to reduce energy consumption, stricter conditions were introduced for the insulation of the building envelope. Increasing the insulation performance of a building envelope is most often achieved by using a layer of high thermal resistance material such as mineral wool. Multi-pane windows with argon or other gas are currently the most popular option for windows. However, the continuous development in building materials brings new solutions. One of them is the VIG (Vacuum Insulating Glass) window technology, which is one of the most advanced technologies allowing for a significant reduction in building energy losses.

The VIG window is not an entirely new technology. It was first described by Zoller in the patent literature in 1913 [6]. Since then, VIG technology has been the subject of a lot of analysis and research [7,8]. Nevertheless, the first one was produced and presented in 1989 by Collins at the ISES Congress. The process of manufacturing the first vacuum plate proved to be very time consuming and demanding [1].

The process of manufacturing VIG plates and introducing them to the market took place in the late 1990s. The production of VIG windows is distinctly different from that of a traditional composite window [6]. Figure 1 shows a schematic of the design of currently used VIG plates.

VIG vacuum glazing consists of two panes of glass hermetically sealed around the edges. The height of the space between the glass plates, where the vacuum is located, is only 0.3 mm. The atmospheric pressure outside the composite panel generates the risk of collapse or contact between the glass panes. To counteract this, a system of support pillars was placed between them [3,9]. The diameter of the pillars is not more than 600 micrometres, and their compressive strength is not less than 400 MPa. Therefore, they are usually made of metal [10].

Two edge sealing methods are currently used. The first one uses a flexible metal edge seal that allows relative movement of the two panes of glass. In the second method, an edge seal is used in which two sheets of glass are rigidly bonded together [11].

Currently, VIG vacuum plates are used in household appliances and building structures. The use of glass as a component of household appliances is justified by its excellent physical and aesthetic properties. In 2018, Siemens, Anthony, and Haier presented a VIG-type perspective window in a refrigerator. The effects of changing the traditional glazing to vacuum glazing were reducing the panel thickness and the energy consumption of a refrigerator [12]. 

VIG windows are characterised by their low thickness. For this reason, they can be the ideal solution for modernisation in historical buildings. In these buildings, the appearance of the windows must fit into the aesthetics of the entire building. Many older buildings have original window frames with single pane glazing. They are thin but have a high heat transfer coefficient. The goal of modernisation of these facilities is to increase energy efficiency while keeping the appearance of the façade intact. VIG plates allow these criteria to be met [12].

Examples of the use of VIG windows during the renovation of historic buildings can be seen in the 17th century Hermitage Museum in Amsterdam, the MIT central campus in Cambridge, or the Prince William V Art Gallery in The Hague [12].

The use of vacuum windows in newly designed buildings is not yet as popular as in retrofitted buildings. However, due to their excellent insulation parameters, they have great potential to also conquer this construction sector [12].

## 2. Materials and Methods

### 2.1. Subject of the Study

In order to characterize different multiple VIG elements, appropriate parameters were adopted. 

Figure 2 presents the diagram of the VIG-type windows’ structure. Following geometric parameters were assumed: *A* and *B* as the VIG dimensions, *t_g_* as the glass panels’ thicknesses, *t_v_* as the vacuum thickness, *w_s_* as the seal width, *n_x_* and *n_y_* as number of pillars in both orthogonal directions, *d_p_* as the support pillars diameter, *x*_0_ and *y*_0_ as the pillars offset from the VIG edges.

The characteristics of three materials used in this study were adopted. Density, Young’s modulus, and Poisson’s ratio for the glass, pillars, and seal were assumed as follows: *ρ_g_*, *E_g_*, *ν_g_*, *ρ_p_*, *E_p_*, *ν_p_*, *ρ_s_*, *E_s_*, *ν_s_*.

### 2.2. Laboratory Experiments

Seven VIG type plates were investigated in this study. The construction of the tested panels is typical. They consist of two sheets of tempered glass hermetically sealed around the edges with a seal made of steel. The thicknesses of the panes are different, and they define the thickness of the whole composite element. The width of the edge seal is 9 mm around the perimeter of the panel. There is a vacuum between the glass panes, and the possible pressure deviation inside the panel is up to 0.1 Pa. The thickness of the vacuum is 0.3 mm. A system of steel support pillars is placed between the glass panes. The diameter of a single pillar is 0.6 mm, and their spacing is 55 mm.

Plates vary in thickness and dimensions in plan view. Table 1 presents the variable geometric parameters of the tested plates and defines the plate numbering, which will be used in this paper. Table 2 presents the fixed parameters of the tested elements.

In order to test the vacuum windows, a special test stand in the shape of a cube with a 2 m side has been created. The structure of the stand is made of 40 mm × 40 mm aluminium profile sections. The composite panels were suspended from the frame structure using fabric elastic bands. It was assumed that the tape would wrap around the slab along all four edges. The tapes used were made of polyester with a width of 25 m. Figure 3 illustrates the method used to suspend the panels.

The basic properties of the elastic tape were determined via additional testing. Four tensile tests were performed on a tape with an initial length of 15 cm using a Zwick Roell Z050 (ZwickRoell, Ulm, Germany) static testing machine. The results of the test are described in Section 2.3. The Young’s modulus of the tape was determined as 4 GPa.

Non-destructive measurement technology based on the PULSE vibration test method from Brüel & Kjær (Nærum, Denmark) were used to test the panels. The basic element in this method is a LAN-XI type 3160 (Brüel & Kjær, Nærum, Denmark) measuring cassette, which is the data acquisition system and the central unit of the entire measurement system. This element has several input and output positions, to which the vibration excitation device, the measuring apparatus, and the computer are connected.

A specialised Brüel & Kjær impact hammer type 8206 with a sensitivity of 22.7 mV/N was used as the excitation device. It allows for generating impulse signals and, contrary to electrodynamic actuators, does not cause additional load of the tested object. The hammer has three interchangeable tips—rubber, plastic, and aluminium. Ultimately, the plastic tip was used because it allowed for accurate reading of results with little risk of overload.

The recording of the dynamic response of the composite structures was measured using Brüel & Kjær type 4399 accelerometers. The accelerometer is used to measure linear and angular acceleration. Three uniaxial accelerometers will be used in this study to study accelerations in the vertical direction. In order not to disturb the structure of the composite plates and the vacuum action, the accelerometers were attached to the surface of the test pieces with beeswax. Two accelerometers were attached to the vacuum glazing-one on the top surface of the plate and one on the bottom surface. Two measurement points were chosen at distances from the edge of the plate equal to 0.2 and 0.5 of the plate’s edge length to minimize the possibility of a situation where the accelerometer is located at a place with zero vibration amplitude. Additionally, a control measurement was made by placing one accelerometer on the aluminium frame. Figure 4, Figure 5 and Figure 6 show the location and mounting of the accelerometers.

A computer with installed software compatible with the measurement equipment and the impact hammer is also connected to the data acquisition system via LAN cable. Figure 7 shows the complete test stand.

Brüel & Kjær software was used to observe and record measurements of the dynamic response of the VIG plates. The Front-end Pulse communicates with the computer via a LAN network cable. However, the computer must know the IP address that the Front-end Pulse uses. The process of setting up the computer to communicate with the Front-end was conducted using the program PULSE Front-end Setup.

PULSE Labshop can transform the measured data into a frequency domain and determine the FRF (Frequency Response Function) and thus modal parameters such as the natural frequency using Fast Fourier Transform (FFT). The Fast Fourier Transform is a powerful algorithm for calculating the Discrete Fourier Transform (DFT). The DFT is a discrete equivalent of the continuous Fourier transform, which is used because of the discrete nature of the measured data. Performing a change of the domain of the function from the domain *x^o^* to the domain *s^o^* for the function *f*(*x*), the Fourier transform can be described by Equation (1):(1)$$\hat{f}(s)=\int_{-\infty }^{+\infty }f\left(x\right)e^{-2\pi ixs}dx$$
in which *i* is an imaginary unit $\left(i^{2}=-1\right)$ [6].

The discrete Fourier transform for a sequence *x_n_* consisting of *N* elements is given by the formula:(2)$$X_{k}=\sum_{n=0}^{N-1}x_{n}e^{-\frac{2\pi i}{N}nk},\;\;\;\;\;\;k\in \left\{0,1,\ldots ,N-1\right\}$$
where $x_{0},\ldots ,\;x_{N-1}$ are the signal samples, and i the imaginary unit $(i^{2}=-1$) [6].

Reducing the number of necessary arithmetic operations in FFT relative to DFT is carried out by dividing the sequence into smaller strings. One of the most popular methods of determining the FFT is the classical Cooley–Turkey algorithm, in which the sequences of signal values have a basis equal to 2. It is assumed that *N* is a power of two and there are *N* samples of the discrete signal. This sequence is first decomposed into strings consisting of samples with even *x_e_*(*n*) and odd *x_o_*(*n*) indices. The Fourier transform of the strings is performed: *X_e_*(*n*), *X_o_*(*n*) and sums the transforms of the whole sequence of samples according to formula:(3)$$X\left(k\right)=X_{p}\left(k\right)+W_{k}^{N}X_{n}(k)$$
(4)$$X\left(k+\frac{N}{2}\right)=X_{p}\left(k\right)-W_{k}^{N}X_{n}(k)$$
(5)$$W_{k}^{N}=e^{-\frac{2\pi jk}{N}}$$
for $0\leq k\leq \frac{N}{2}$ [13].

The signals connected to the data acquisition system were added in PULSE Labshop. The first three signals were accelerometers, located on the top and bottom surfaces of the VIG plate and aluminium frame, respectively, acc_top, acc_bottom, and acc_frame. The fourth and last input added was a vibration forcing device, the impact hammer: hammer. A signal group was also created to which all transducers were added. This group was connected to the FFT analyser. Then, cross-spectrum functions were added: cross-spectrum(acc_top,hammer)–Inputcross-spectrum(acc_bottom,hammer)–Inputcross-spectrum(acc_top,acc_frame)–Inputcross-spectrum(acc_bottom,acc_frame)–Input

After all settings had been applied, measurements were made using the Start button. Five combinations of the VIG plate test method were considered, differing in the position of the accelerometers on the plate and where the loads were applied. Two accelerometer location positions were assumed, location No. 1 and location No. 2 (Figure 6). The forcing will be applied at 2 selected points and at random. Three trials were performed for each combination. Table 3 shows the description of the combinations and their numbering, which will be used later in the paper.

### 2.3. Finite Element Method 

In this study, the finite element method (FEM) software (ABAQUS CAE 2017) was used in order to perform a dynamic analysis. FEM allowed for the creation of a model with full 3D representation of support pillars and, consequently, provided greater accuracy [14,15]. The adopted model allows the inclusion of stress concentration in support pillars and shear deformation. It should be mentioned that vacuum thickness compared to glass pane thickness is small; therefore, 3D finite elements provide more accurate results when it comes to dynamic response.

Calculations in ABAQUS/CAE 2017 were performed using PLGrid Infrastructure, which provides access to clusters located in five High Performance Computing centres. Computational tasks are outsourced through a so-called middleware, which manages the resources of all hubs centrally. The WCSS hub was used in this analysis. CPU run time depended on model size and was about 4–30 min.

In this paper, C3D8R was assumed (Figure 8)—first order, reduced integration element with hourglass control activated.

An interesting study which describes numerical difficulties of shear locking and hourglassing may be investigated in paper [17]. 

The interpolation function ***u*** for C3D8R element is given by Equation (6):(6)$$u=N^{I}\left(g,h,r\right)u^{I}\;\mathrm{sum}\;\mathrm{on}\;I.$$
where *N^I^* are isoparametric shape functions, and *I* is the node of the element.

The shape functions are the same as for the C3D8 element and can be found in [16,18] (7):(7)$$N^{I}\left(g,h,r\right)=\frac{1}{8}\Sigma^{I}+\frac{1}{4}g\Lambda_{1}^{I}+\frac{1}{4}h\Lambda_{2}^{I}+\frac{1}{4}r\Lambda_{3}^{I}+\frac{1}{2}hr\Gamma_{1}^{I}+\frac{1}{2}gr\Gamma_{2}^{I}+\frac{1}{2}gh\Gamma_{3}^{I}+\frac{1}{2}ghr\Gamma_{4}^{I}$$
where *I* denotes the node of the element. The last four vectors $\Gamma_{\alpha }^{I}$, α=1,2,3,4 are the hourglass base vectors. The gradient matrix $B^{I}$ is defined by integrating over the element:(8)$$B_{i}^{I}=\frac{1}{V_{el}}\int_{V_{el}}^{}N_{i}^{I}\left(g,h,r\right)dV_{el,}\;\;N_{i}^{I}\left(g,h,r\right)=\frac{\partial N^{I}}{\partial x_{i}}$$
where $V_{el}$ is volume of the element and *i =* 1, 2, 3. In the centroidal strain formulation, the gradient matrix is simply given:(9)$$B_{i}^{I}=N_{i}^{I}\left(0,0,0\right)$$

The model used in this study is precisely described in [5].

The dynamic analysis of VIG plate has been performed in this paper. To perform a dynamic analysis, the linear perturbation step was created. The frequency extraction performs eigenvalue extraction to calculate the natural frequencies and the corresponding mode shapes of a system. Geometric linearity was accounted for. The eigenvalue problem for the natural frequencies was considered:(10)$$\left(-\omega^{2}M^{MN}+K^{MN}\right)\phi^{N}=0$$
where $M^{MN}$ is the mass matrix, $K^{MN}$ is the stiffness matrix, $\phi^{N}$ is the eigenvector (mode of vibration), and *M* and *N* are degrees of freedom. The Lanczos method with 30 eigenvalues set was chosen.

The static scheme of the considered VIG plate was assumed in accordance with the experimental analysis. Boundary conditions were represented in the created FEM model as eight small shell elements on short sections of glass panes’ edges, close to the corners of the VIG element, rotated by the appropriate angle in order to imitate elastic bands (Figure 9). 

This tiny elements’ stiffness is significantly higher in comparison to glass panes’ stiffness. Their main goal was to imitate the angle of the bands. This high stiffness elements are restrained by two one-directional springs. The spring’s stiffness is the total linear stiffness of one band split into two nodal springs (11):(11)$$k_{z}=\frac{E\cdot A}{L_{b}}\cdot \frac{w_{b}}{w_{b}}\cdot 0.5=\frac{E\cdot A}{L_{b}}\cdot 0.5$$
where $k_{z}$ is nodal stiffness, E is a band’s modulus of elasticity, $L_{b}$ is a length of band A is a band’s cross-section area, and $w_{b}$ is a band’s width.

In order to determine the Young’s modulus of the bands, experimental test were performed. Four samples were subjected to a uniaxial tensile test (Figure 10) and the proper results were obtained.

The samples’ length was 150 mm, width 25 mm, and thickness 0.75 mm. The force-displacement relationship obtained from the experiments is presented in Figure 11.

Therefore, based on Figure 11, the band’s modulus of elasticity was assumed as 4.0 GPa. 

### 2.4. Machine Learning Method

#### 2.4.1. Extreme Gradient Boosting

The Extreme Gradient Boosting method is one of many machine learning (ML) algorithms. It is a Gradient Boosting Decision Trees (GBDT) method. It provides parallel tree boosting and is the leading machine learning library for regression, classification, prediction, and ranking problems. This method is based on supervised machine learning, decision trees, ensemble learning, gradient boosting.

The XGB method was initially a research project by Tianqi Chen [19] within a research group called Distributed (Deep) Machine Learning Community (DMLC). It should also be mentioned that in 2019 the XGB method was one of the winners of the prestigious InfoWorld Technology of the Year award. The XGB library is constantly being developed around the world due to the ever-increasing number of scientists contributing to it. XGBoost machine learning models perfectly combine performance and processing time in relation to other algorithms.

A detailed description of the XGB algorithm structure is presented below [20].

Data set was assumed as follows:(12)$$D=\left\{\left(x_{i},y_{i}\right)\right\};i=1,2,\ldots ,n$$

The model that was used for training and learning consists of *k* trees. The result obtained using this model can be described as follows:(13)$$\hat{y}=\emptyset \left(x_{i}\right)=\sum_{k=1}^{K}f_{k}(x_{i});f_{k}\in F$$
where F is a hypothetical space and $f\left(x\right)$ is a regression decision tree:(14)$$F=\{f\left(x\right)=\omega_{q\left(x\right)}\}$$

In Equation (14), $q\left(x\right)$ is the leaf (the end node of the decision tree of the *x*-th sample) and ω is the result of this leaf. The predicted result for the *t*-th iteration can be determined as follows:(15)$${\hat{y}}_{i}^{t}={\hat{y}}_{i}^{t-1}+f_{t}\left(x_{i}\right)$$

Therefore, the objective function can be defined as:(16)$$J\left(f_{t}\right)=\sum_{i=1}^{n}L\left(y_{i},{\hat{y}}_{i}^{t-1}+f_{t}\left(x_{i}\right)\right)+Ω\left(f_{t}\right)$$
where L is the error function and $Ω\left(f_{t}\right)$ represents the complexity of the model.

T is the total number of leaves and ω is the result of the leaf:(17)$$Ω\left(f_{t}\right)=\gamma T_{t}+\lambda \frac{1}{2}\sum_{j=1}^{T}\omega_{j}^{2}$$

Then, Equation (17) can be simplified by applying the Taylor expansion of the second order:(18)$$J\left(f_{t}\right)=\sum_{i=1}^{n}\left[L\left(y_{i},{\hat{y}}_{i}^{t-1}\right)+g_{i}f_{t}\left(x_{i}\right)+\frac{1}{2}h_{i}f_{t}^{2}\left(x_{i}\right)\right]+Ω\left(f_{t}\right)$$
(19)$$g_{i}=\frac{\partial L\left(y_{i},{\hat{y}}_{i}^{t-1}\right)}{\partial {\hat{y}}_{i}^{t-1}}$$
(20)$$h_{i}=\frac{\partial^{2}L\left(y_{i},{\hat{y}}_{i}^{t-1}\right)}{\partial {{\hat{y}}_{i}^{t-1}}^{2}}$$

Based on the above analysis, the final objective function can be written as:(21)$$J\left(f_{t}\right)=\sum_{i=1}^{n}\left[g_{i}\omega_{q(x_{i})}+\frac{1}{2}h_{i}\omega_{q(x_{i})}^{2}\right]+\gamma T_{t}+\lambda \frac{1}{2}\sum_{j=1}^{T}\omega_{j}^{2}$$

In summary, the objective function is optimised and the optimal solution is as follows:(22)$$\omega_{j}^{*}=-\frac{\sum_{i\in I_{j}}g_{i}}{\sum_{i\in I_{j}}h_{i}+\lambda }$$
(23)$$J\left(f_{t}\right)=-\frac{1}{2}\sum_{j=1}^{T}\frac{{\left(\sum_{i\in I_{j}}g_{i}\right)}^{2}}{\sum_{i\in I_{j}}h_{i}+\lambda }+\gamma T$$

In order to determine the effectiveness of the XGB model, the value of the root mean square error (*RMSE*) was determined. The equation for the described expression is as follows:(24)$$RMSE=\sqrt{\frac{1}{n}\sum_{i=1}^{n}{\left(y^{\left(i\right)}-\bar{y}\right)}^{2}}$$
where $y^{\left(i\right)}$ is the target value used to train the model and $\bar{y}$ is the value predicted by the model.

#### 2.4.2. Deep Neural Networks

Neural Networks (NN) are part of the ML and are also the basis of Deep Learning (DL) algorithms. Their goal is to compute the structure of the biological networks of neurons in the human brain. They consist of an input layer, one or more hidden layers and an output layer. Each artificial neuron of a given layer connects to each artificial neuron of adjacent layers (or layers) and has an associated weight and activation threshold. If a neural network consists of a minimum of four layers-an input layer, an output layer, and at least two hidden layers–then it can be called a Deep Neural Network (DNN) (Figure 12).

The first step towards creating something like neural networks in the human brain was a study published in 1943 [21]. McCulloch and Pitts compared it with binary threshold neurons to Boolean logic.

In 1958, Frank Rosenblatt took another step beyond the work of McCulloch and Pitts [21] by adding a weight equation to the neuron. He created and described the concept of a perceptron, and then used it to create an algorithm, whose purpose was to distinguish the cards marked on the left from the cards marked on the right [22].

NN algorithms are also used in engineering in order to obtain soil geotechnical properties or mechanical properties of surface layered systems based on non-destructive experimental test [23].

With time, there were more and more studies on neural networks. Their use has grown steadily and continues to grow. Despite the extensive use of neural networks in many aspects of everyday life, this topic is still being developed and improved, and its final goal is to achieve the effectiveness of the human brain.

In this paper, a deep neural network with back propagation was used, which can be classified as Multilayer Perceptron (MLP) networks. The description of this algorithm is presented below [24].

The following Equation (25) describes the nonlinear neuron used:(25)y=φ(e)
where φ is the selected non-linear activation function. The signal e describes the total excitation of the neuron (26):(26)$$e=\sum_{i=0}^{n}w_{i}x_{i}$$
where signals $<x_{1},x_{2},\ldots ,x_{n}>$ are components of the vector X, while $w_{0}$ is a constant component of the neuron, therefore $x_{0}\equiv 1$.

Assuming W as the vector $<w_{0},w_{1},w_{2},\ldots ,w_{n}>$, the signal e in the vector form can also be written as follows (27):(27)$$e=W^{T}\;X.$$

For a multi-layer network, a training sequence of the following structure can be assumed (28):(28)$$U=<<X^{\left(1\right)},Z^{\left(1\right)}>,<X^{\left(2\right)},Z^{\left(2\right)}>,\ldots ,<X^{\left(M\right)},Z^{\left(M\right)}>>.$$

It consists of pairs in the form of $<X^{\left(j\right)},Z^{\left(j\right)}>$ which contain *n*-dimensional X vectors determined in the *j*-th step of the learning process and *k*-dimensional Z sets defining the outputs from the elements closing the network in this step. If the vector $Y^{\left(j\right)}$ of the output signals from these elements does not meet the requirements, i.e., an error determined by the difference $Z^{\left(j\right)}-Y^{\left(j\right)}$ is specified, then the determination of the layer of neurons responsible for this error it will not be possible. The solution to this problem is to use the backpropagation error algorithm. It means that having the determined error $\delta^{\left(m)(j\right)}$ (occurring during the implementation of the *j*-th step of the learning process in the neuron number *m*), it is possible to transfer this error back to all neurons whose signals were inputs for the *m*-th neuron.

Due to the lack of possibility for a strict separation of input and output signals (the output signals of one layer become input signals of the next layer), a uniform numbering of neurons was adopted (without dividing them into input, output, and hidden layers) and the constant $y_{m}^{\left(j\right)}$ for the signal appearing at the output of the *m*-th neuron in the *j*-th step of the learning process.

Moreover, it should be noted that by introducing a uniform designation, also for the input signals, each input of the network can be defined as buffered by a neuron whose task is to convert the input signal $x_{i}^{\left(j\right)}\in X^{\left(j\right)}$ into the signal $y_{m}^{\left(j\right)}$ (the neuron is linear and the weight value of the single input of this neuron is 1). This relationship is determined by the following Equation (29):(29)$$y_{m}^{\left(j\right)}=x_{i}^{\left(j\right)}.$$

It can be concluded that the only way to distinguish the type of a given signal (input, output, or one of the hidden layer signals) is to determine the number *m*. Therefore, the sets of neuron numbers $M\left(\forall_{m}\in M\right)$ of the following subsets were introduced:

$M_{x}$—set of numbers of input neurons to the network;

$M_{y}$—set of numbers of output neurons from the network;

$M_{u}$—a set of neuron numbers of the hidden layer of the network;

$M_{i}$—a set of numbers of neurons providing input signals to a specific neuron currently under consideration;

$M_{o}$—a set of neuron numbers to which the given, currently considered neuron, sends its output signal.

Using the above markings, it is possible to describe the output signal of each *m*-th neuron of the network in the *j*-th step of the learning process (30):(30)$$y_{m}^{\left(j\right)}=\varphi \left(\sum_{i\in M_{i}}w_{i}^{\left(m\right)\left(j\right)}y_{i}^{\left(j\right)}\right).$$
where $w_{i}^{\left(m\right)\left(j\right)}$ is the value of the synapse weighting factor (connection of two neurons) connecting the input of the m-th neuron with the output of the i-th neuron during the j-th step of the learning process. The order of calculations is directly related to the sequence of transmitting signals from the network input to the neurons in subsequent layers. First, determining $y_{m}^{\left(j\right)}$ for $m\in M_{x}$, then for $m\in M_{u}$ (successively in all layers from the entrance towards the exit), and finally for $m\in M_{y}$.

In the case of the backpropagation method, the proper order to find the values of the corrected synaptic weights is exactly the opposite of that required to determine the values of the signals in successive network elements.

The first step is to determine the correction for the neurons constituting the initial network layer ($m\in M_{y}$). Since the expected values of $z_{m}^{\left(j\right)}$ exist in the training sequence for individual signals $y_{m}^{\left(j\right)}$, the determination of the error $\delta_{m}^{\left(j\right)}$ can be carried out directly. It is necessary to assume that the numbering of the vector components $Z^{\left(j\right)}$ is the same as the numbering of the neurons that make up the output layer of the network. Then:(31)$$\Delta w_{i}^{\left(m\right)(j)}=\eta \delta_{m}^{\left(j\right)}\frac{d\varphi \left(e\right)}{de_{m}^{\left(j\right)}}y_{i}^{\left(j\right)},$$
where for $m\in M_{y}$:(32)$$\delta_{m}^{\left(j\right)}=z_{m}^{\left(j\right)}-y_{m}^{\left(j\right)}.$$

This Equation (32) for $m\in M_{y}$ and for φ(e) in the form of a logistic function can be written as follows (33):(33)$$\Delta w_{i}^{\left(m\right)(j)}=\eta \left(z_{m}^{\left(j\right)}-y_{m}^{\left(j\right)}\right)\left(1-y_{m}^{\left(j\right)}\right)y_{i}^{\left(j\right)}y_{m}^{\left(j\right)}.$$

Using an analogy, the following rule can be defined for neurons of hidden layers (34):(34)$$\Delta w_{i}^{\left(m\right)(j)}=\eta \delta_{m}^{\left(j\right)}\frac{d\varphi \left(e\right)}{de_{m}^{\left(j\right)}}y_{i}^{\left(j\right)},$$

Nonetheless, for $m\in M_{u}$, it is not possible to directly determine the value of $\delta_{m}^{\left(j\right)}$.

Assuming that $M_{o}\subseteq M_{y}$ (although the considered neuron belongs to the hidden layer, its signal reaches only output layer neurons, i.e., those for which the error values $\delta_{m}^{\left(j\right)}$ can be easily determined), it can be shown [25] that the error $\delta_{m}^{\left(j\right)}$ of the hidden layer neuron can be determined by backpropagation of the errors detected in the signal receiving layer:(35)$$\delta_{m}^{\left(j\right)}=\sum_{k\in M_{o}}w_{m}^{\left(k)(j\right)}\delta_{k}^{\left(j\right)},$$
where the factor $w_{m}^{\left(k)(j\right)}$ is the weight in the neuron number k at its input number m, i.e., receiving a signal from the currently considered neuron. This means that the back-projected errors are multiplied by the same coefficients by which the transmitted signals were multiplied, only that the direction of information transmission was reversed in this case.

In order to determine the effectiveness of the NN model, the value of the root mean square error (RMSE) was determined (24).

#### 2.4.3. Input and Output Data

The key aspect of creating an effective predictive model is the appropriate selection of the data set in order to conduct the learning process of the model. It is important due to the goal of the algorithm, which is finding a predictable pattern that is hidden in the entire data set.

The purpose of splitting data is to prevent undesirable extrapolation (building a model from data that should not be known), overfitting (the process of designing a model that conforms so closely to a provided data set that it becomes ineffective in the future predictions), and underfitting (the process of designing a model that conforms so loosely to a provided data set that it becomes ineffective in the future).

The assumed data sets include different types of parameters. These are both scalar values with and without units. The data, on which models based on ML algorithms could be trained, must be unitless. Table 4 presents all the parameters of input and output data assumed in this paper.

Precise description of the data set that has been obtained from numerical analysis is discussed in Section 3.2. The entire data set has been randomly divided into a training data set (80%) and a test data set (20%).

Using the described data sets, two different algorithms were developed–XGB and DNN. The reason was the desire to examine the performance of these methods in reference to the considered problem.

## 3. Results

### 3.1. Experimental Analysis

As a result of the experimental study, frequency spectrum plots were obtained from which the natural frequencies of the plates can be read. Figure 13, Figure 14, Figure 15, Figure 16, Figure 17, Figure 18, Figure 19 and Figure 20 show the graphs for a 500 mm × 600 mm × 8.3 mm plate.

The natural frequencies were determined by looking for local extremes on the characteristics of the spectra. It was also taken into account that the function is not increasing and decreasing in a smooth manner. From the spectra plots, the ranges in which local extremes are expected were determined and the maximum values were chosen.

In order to correctly analyse these results, numerical models (Section 2.3) for the geometrical and mechanical parameters of the tested VIG-type windows (Table 1 and Table 2) were created and natural frequencies obtained. Numerically obtained mode shapes were taken into account. This is very important in the case of 1000 mm × 1000 mm plates because the natural frequencies for some two consecutive mode shapes are the same. This is due to the same geometrical parameters in two directions.

The averaged values of natural frequencies obtained from all measurements are shown in Table 5.

### 3.2. Numerical Analysis

The eigenvalues (natural frequencies) for a given set of parameters were obtained using the numerical models described in Section 2.3. Mesh density assumption has been analysed in the same way as described in the paper [5]. Table 6 shows all the values adopted for the implementation of the numerical models on the basis of which the values of the natural frequencies were determined. In total, 315 different numerical models were created by making combinations that include all the given values.

First thirty natural frequencies were determined. Mode shapes for the first ten natural frequencies are shown in Figure 21. The greatest relative deflection is marked in red.

### 3.3. Machine Learning

#### 3.3.1. Extreme Gradient Boosting

In order to create XGB algorithm in Python XGBRegressor function was used. Linear regression was assumed for this model. Moreover, to find the most optimal parameters of the predictive model, multiple combinations of models’ parameters were analysed. These parameters are shown in Table 7. Final parameters were chosen to obtain the least possible Root Mean Squared Error (RMSE) that was determined by Equation (24).

The errors, obtained for the assumed parameters of the created Extreme Gradient Boosting predictive model, are presented in Table 8.

Figure 22 shows one random tree from the whole set of automatically created decision trees.

Figure 23 presents feature importance diagram for the obtained results from the XGB predictive model.

#### 3.3.2. Neural Networks

In order to create DNN script in Python, pytorch, skorch, and sklearn libraries were used. Moreover, to find the most optimal parameters of NN, GridSearch function from sklearn library was taken. These parameters are shown in Table 9. Final parameters were chosen to obtain the least possible Root Mean Squared Error (RMSE) that is determined by Equation (24).

It has to be mentioned that all the data in the provided data set had to be normalised in order to train DNN algorithm; this is due to a low effectiveness of neural networks when it comes to values with different a order of magnitude. Therefore, the normalizer function from sklearn library was used.

The errors, obtained for the assumed parameters of the created Deep Neural Network predictive model, are presented in Table 10.

## 4. Machine Learning in Non-Destructive Experiments of VIG Units

The final part of this paper is to use the created predictive models in order to obtain the Young’s modulus of pillars located between glass panes of the tested VIG plates. In this case, the predictive models will be used as a function that determines the Young’s modulus value, and the experimental results will be used as variables for the mentioned function (Table 5). Based on these assumptions, the wanted values were determined using two predictive models for seven tested VIG elements. The results are presented in Table 11.

## 5. Conclusions and Future Research

The aim of this study was to anticipate Young’s modulus of VIG pillars using ML predictive models. Two different types of ML models were investigated–Extreme Gradient Boosting and Deep Neural Networks. 

Summarizing briefly all the data obtained and presented, the following could be stated:the experimental tests were properly performed;the created numerical model is extremely sensitive with respect to natural frequencies;the provided data are immensely corelated and ML models are not able to generalize themselves;the final results are not satisfactory.

Each issue is described further in this section.

In accordance with the results obtained from the experimental tests, it could be concluded that the number of combinations, concerning different locations of accelerometers and different locations of force application, was huge enough to provide satisfactory results, because all the wanted natural frequencies (obtained from the numerical models) were identified. In order to obtain more than the first thirty natural frequencies, more combinations should be investigated. Moreover, the performed number of trials was also enough in order to obtain proper results, because the results from those trials were almost the same.

The numerical model, based on the assumptions from [5], provides accurate results because it does not omit a problem with small ratio of distance between glass panes (thickness of vacuum layer) in comparison to glass panes’ thicknesses. It should be also mentioned that in [5], authors presented that changing the value of pillars’ modulus of elasticity do not significantly influence the dynamic properties (natural frequencies) of VIG plates. Therefore, the sets of data with natural frequencies, provided for the VIG window with the same overall dimensions, are highly similar to each other.

Considering provided data, despite rather low relative errors presented in Table 11, the results should have definitely lower error. Despite the dataset being rather tiny, it could be observed that these data are highly corelated (Figure 24). Although the results from XGB algorithm are slightly more promising than results from DNN algorithm, both predictive models are unable to generalize themselves. Predicted values are close to the mean value of the whole set. This is probably due to two following aspects: pillars’ influence on natural frequencies is not significant and provided data could not be highly diverse. The assumptions for the data sets adopted this study are insufficient. 

There are two possible solutions to solve this problem. The first one is to highly extend the data set. More complex data could cause definitely lower correlation of variables between training and test data sets. Instead of using the natural frequencies, the complete spectra of the dynamic response could be used. The second possibility is to use meta-learning [24]. It refers to learning algorithms that learn from other learning algorithms and became more powerful. It is also related to model selecting and algorithm tuning. 

Considering the fact, that non-destructive tests of VIG plates could play a key role in testing how these windows are aging, further investigation have to be performed. Therefore, future research will continue to focus on the development and validation of two mentioned methods.

## Figures and Tables

**Figure 1 materials-16-05055-f001:**
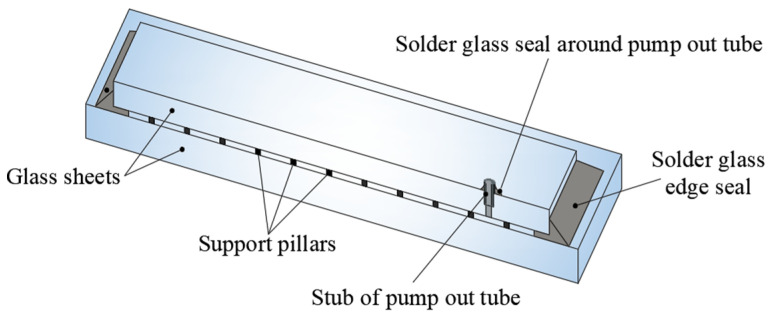
Scheme of construction of the VIG unit.

**Figure 2 materials-16-05055-f002:**
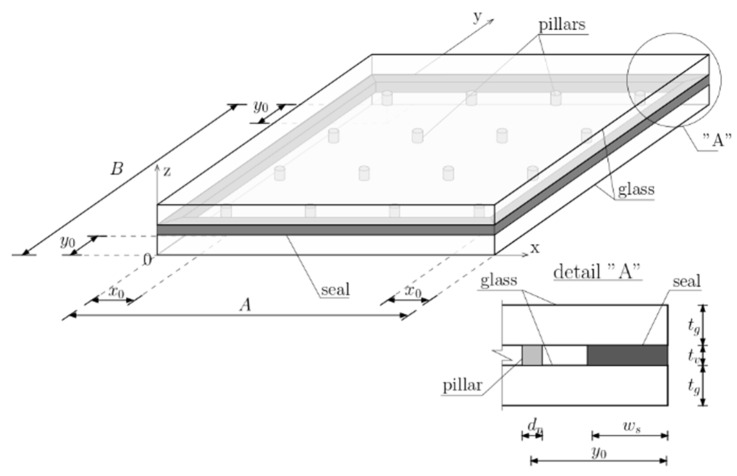
VIG panel structure scheme.

**Figure 3 materials-16-05055-f003:**
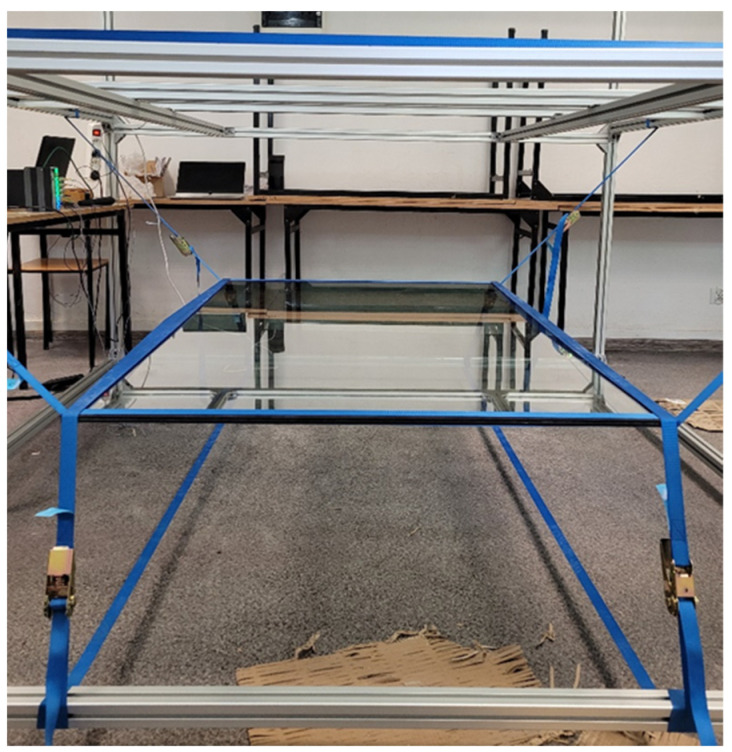
Board suspension method using elastic bands.

**Figure 4 materials-16-05055-f004:**
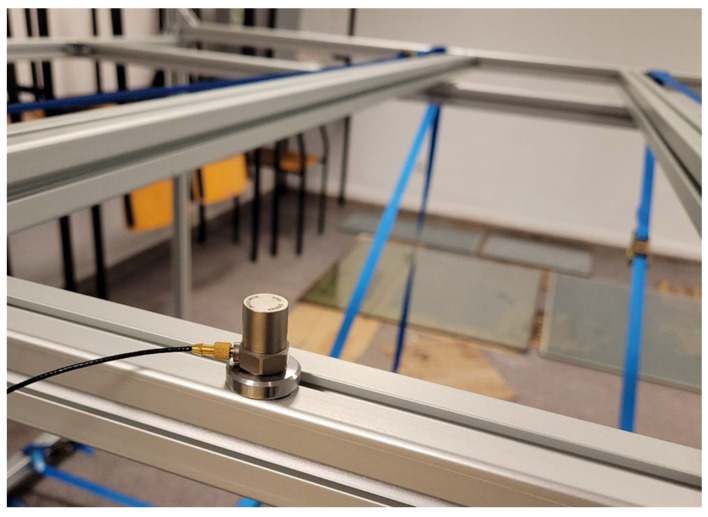
Mounting method and location of accelerometer on aluminium frame.

**Figure 5 materials-16-05055-f005:**
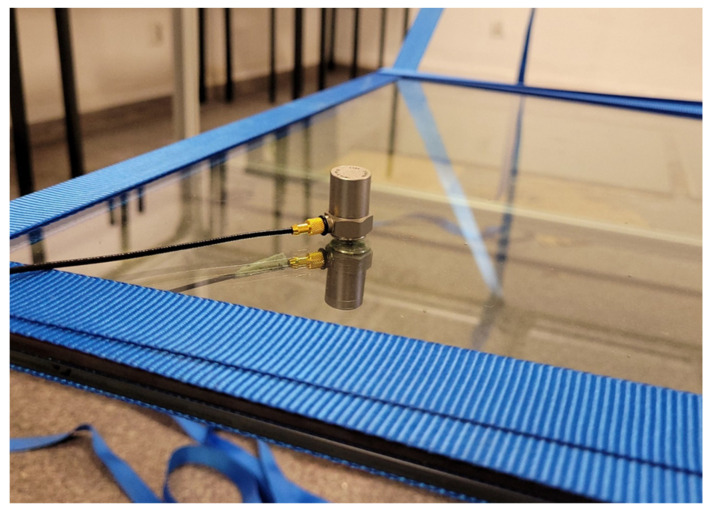
Mounting method of accelerometers on the VIG panel.

**Figure 6 materials-16-05055-f006:**
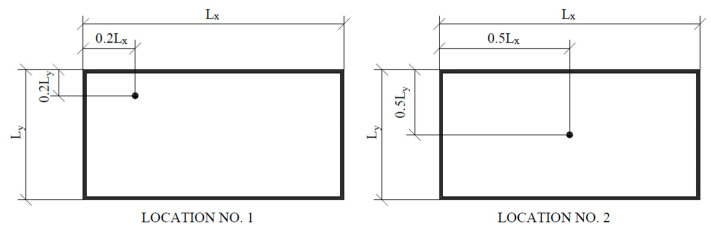
Accelerometer locations on the VIG panel.

**Figure 7 materials-16-05055-f007:**
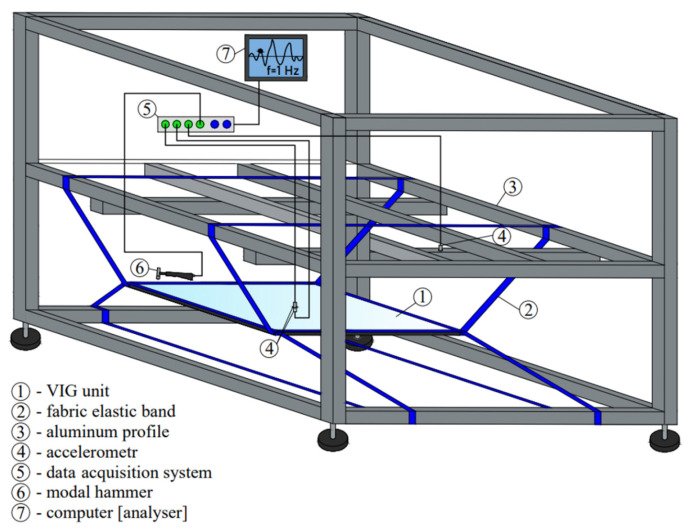
Scheme of the test stand with its elements.

**Figure 8 materials-16-05055-f008:**
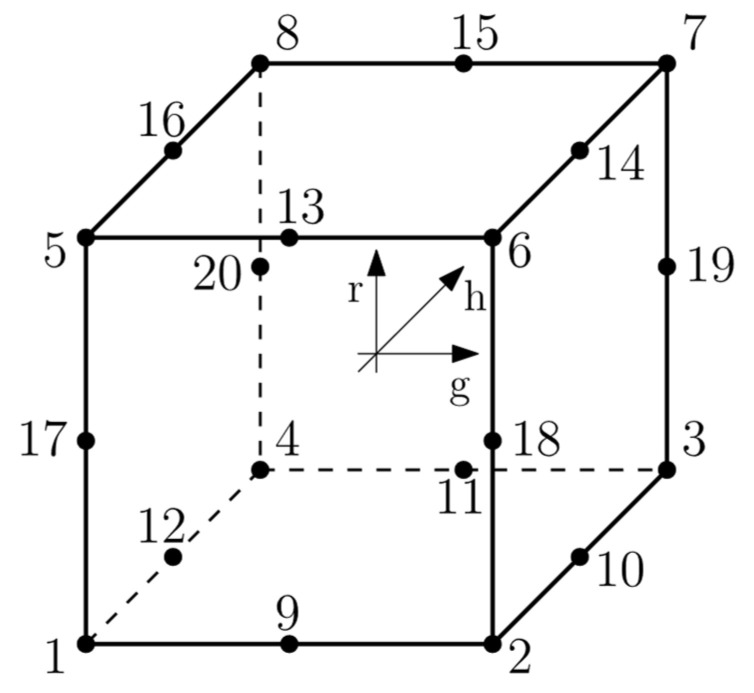
C3D8R element designations where numbers indicates number of nodes [16].

**Figure 9 materials-16-05055-f009:**
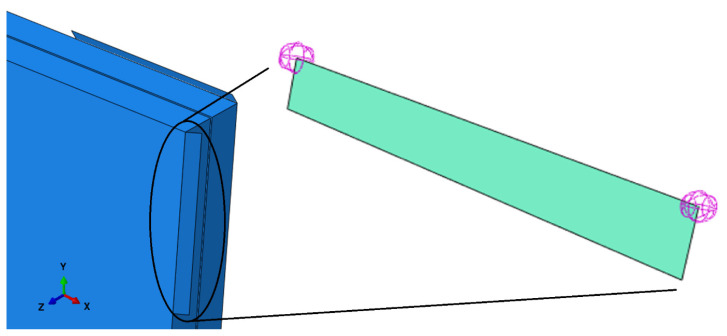
Numerical imitation of elastic bands in ABAQUS.

**Figure 10 materials-16-05055-f010:**
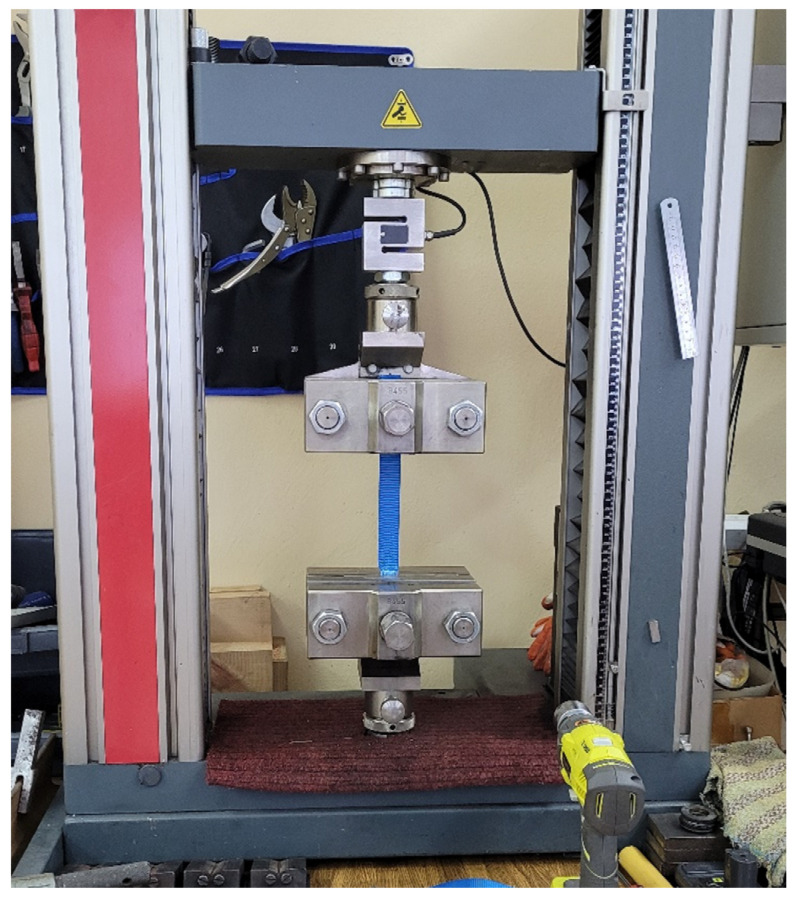
Experimental tests of support bands.

**Figure 11 materials-16-05055-f011:**
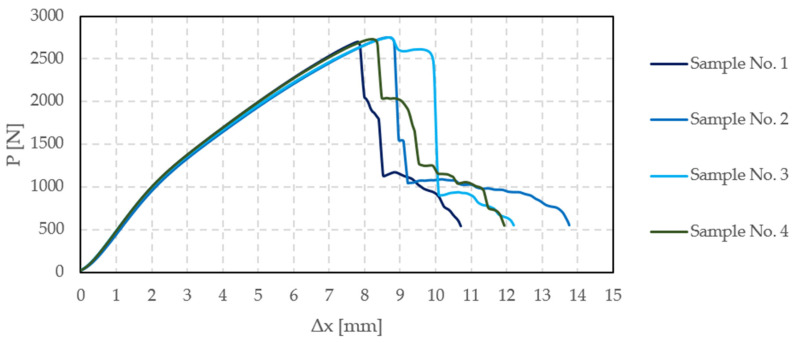
Force-displacement relationship for the four experimentally tested bands.

**Figure 12 materials-16-05055-f012:**
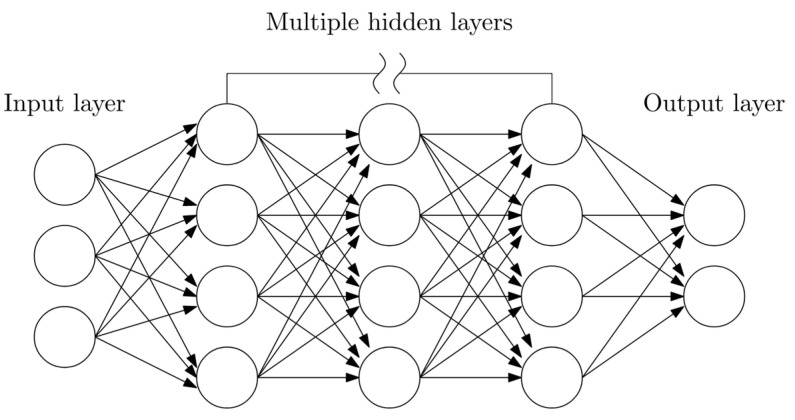
Deep Neural Networks diagram.

**Figure 13 materials-16-05055-f013:**
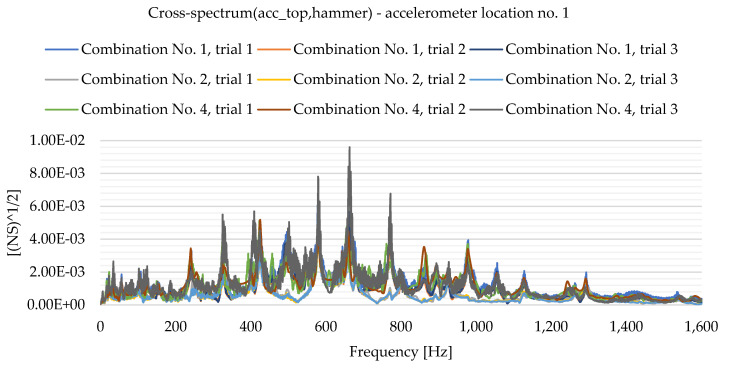
Frequency spectrum diagram (acc_top,hammer)—accelerometer location No. 1.

**Figure 14 materials-16-05055-f014:**
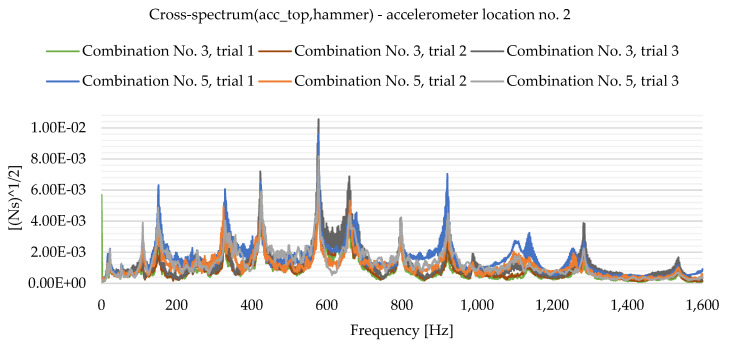
Frequency spectrum diagram (acc_top,hammer)—accelerometer location No. 2.

**Figure 15 materials-16-05055-f015:**
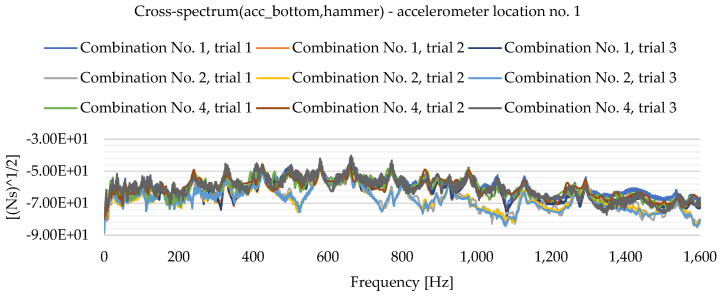
Frequency spectrum diagram (acc_bottom,hammer)—accelerometer location No. 1.

**Figure 16 materials-16-05055-f016:**
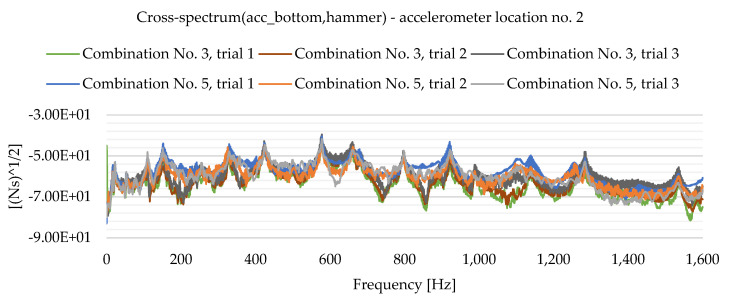
Frequency spectrum diagram (acc_bottom,hammer)—accelerometer location No. 2.

**Figure 17 materials-16-05055-f017:**
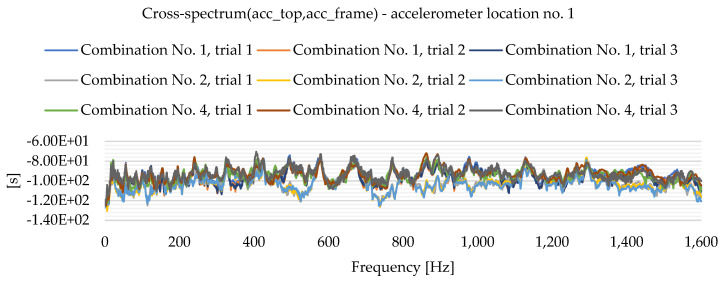
Frequency spectrum diagram (acc_top,acc_frame)—accelerometer location No. 1.

**Figure 18 materials-16-05055-f018:**
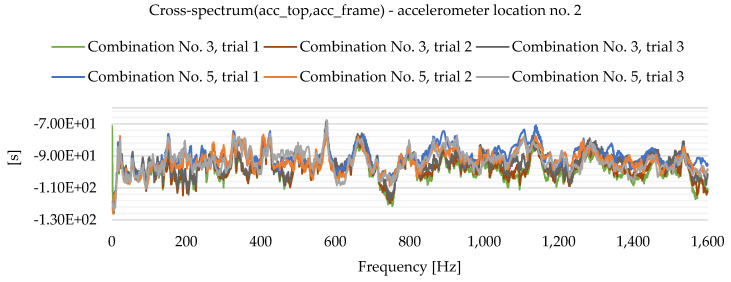
Frequency spectrum diagram (acc_top,acc_frame)—accelerometer location No. 2.

**Figure 19 materials-16-05055-f019:**
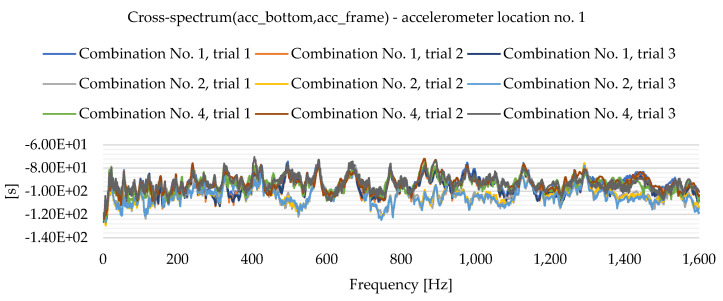
Frequency spectrum diagram (acc_bottom,acc_frame)—accelerometer location No. 1.

**Figure 20 materials-16-05055-f020:**
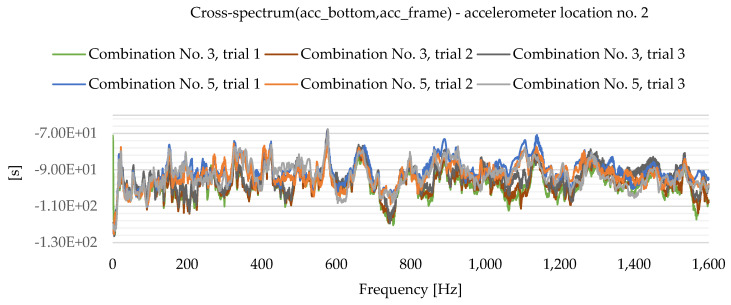
Frequency spectrum diagram (acc_bottom,acc_frame)—accelerometer location No. 2.

**Figure 21 materials-16-05055-f021:**
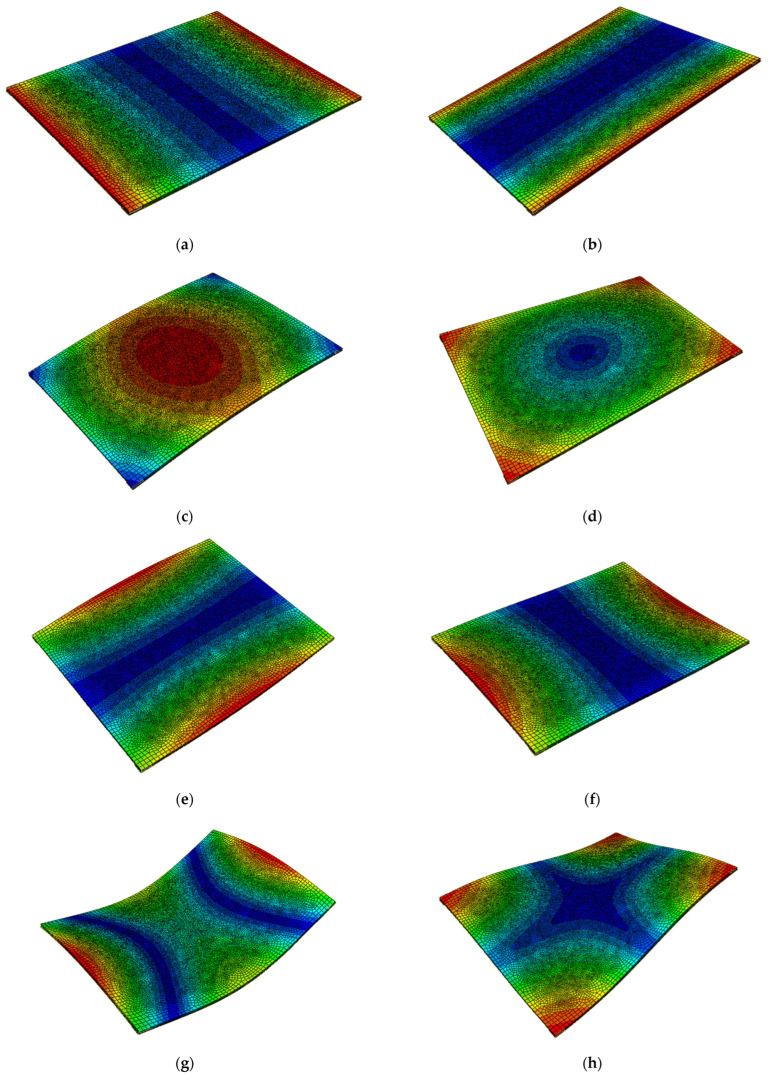
Mode shapes for the first ten natural frequencies from (**a**–**j**).

**Figure 22 materials-16-05055-f022:**
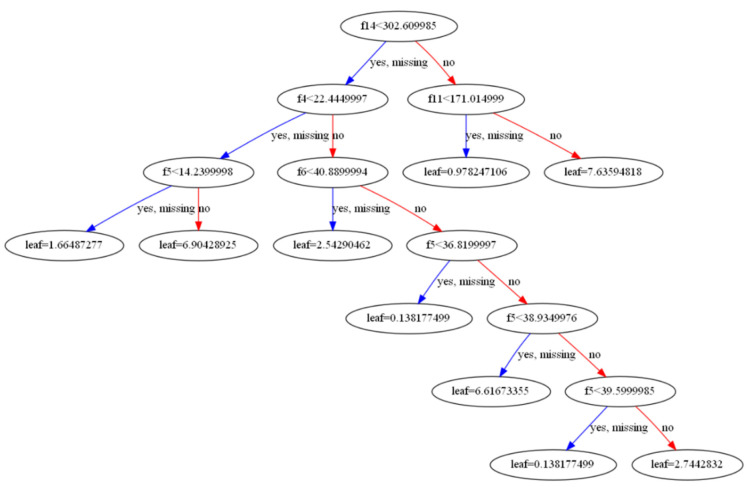
One random XGB decision tree generated by the algorithm.

**Figure 23 materials-16-05055-f023:**
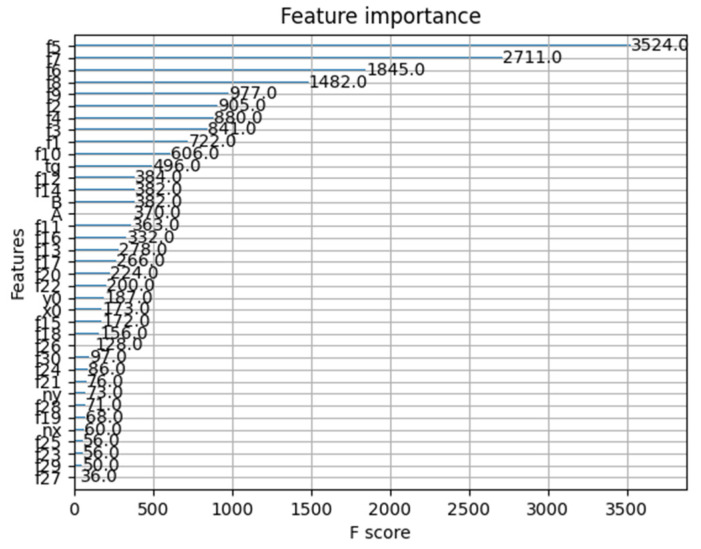
Feature importance diagram for the generated XGB model.

**Figure 24 materials-16-05055-f024:**
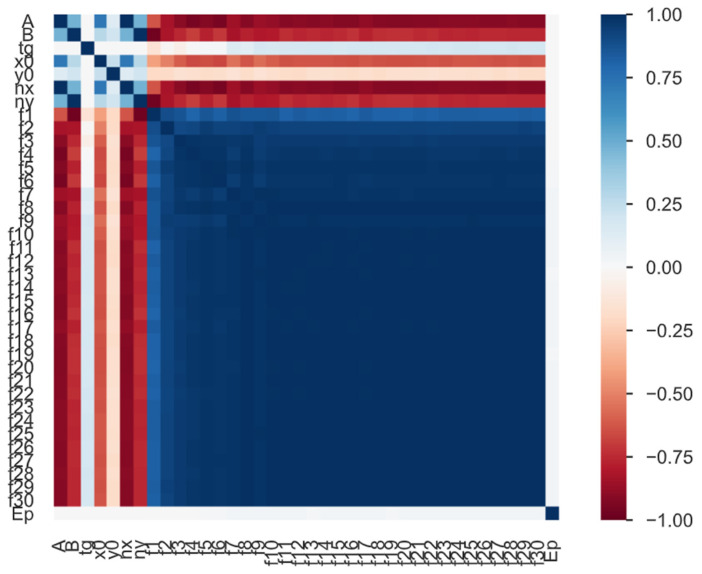
The Spearman’s rank correlation coefficient for provided dataset.

**Table 1 materials-16-05055-t001:** Variable geometric parameters of the tested VIG plates.

No.	Dimensions in Projection	Overall Thickness	Thickness of a Single Glass Pane	Number of Pillars along the Shorter Side	Number of Pillars along the Longer Side
	[mm × mm]	[mm]	[mm]	[-]	[-]
1	400 × 800	10.3	4	8	10
2	500 × 600	8.3	4	8	10
3	500 × 600	10.3	5	6	14
4	1000 × 1000	8.3	4	17	17
5	1000 × 1000	10.3	5	17	17
6	1000 × 1000	12.3	6	17	17
7	1000 × 1500	12.3	6	26	17

**Table 2 materials-16-05055-t002:** Constant parameters of the tested VIG plates.

Plate Element	Material	Thickness	Height	Diameter	Spacing
		[mm]	[mm]	[mm]	[mm]
Glass sheets	Tempered Glass: Poisson’s ratio: *υ_g_* = 0.22 Density: *ρ_g_* = 2500 kg/m^3^ Young’s modulus: *E_g_* = 72 GPa	according to Table 1	n.a.	n.a.	n.a.
Gas between the sheets of glass	Vacuum (0.1 Pa)	0.3	n.a.	n.a.	n.a.
Edge sealing	Steel Poisson’s ratio: *υ_u_* = 0.31 Density: *ρ_u_* = 7850 kg/m^3^ Young’s modulus: *E_u_* = 200 GPa	n.a.	n.a.	n.a.	n.a.
Supporting pillars	Steel Poisson’s ratio: *υ_s_* = 0.31 Density: *ρ_s_* = 7850 kg/m^3^ Young’s modulus: *E_s_* = 200 GPa	n.a.	0.3	0.6	55

**Table 3 materials-16-05055-t003:** Combinations of VIG plate test method.

Combination No.	Accelerometer Location	Place of Application of Force
1	Location No. 1	Location No. 1 on the opposite corner
2	Location No. 2	Location No. 1
3	Location No. 1	Location No. 2
4	Location No. 1	Random
5	Location No. 2	Random

**Table 4 materials-16-05055-t004:** Parameters of input and output data.

Parameter	Unit	Dataset Name	Description
INPUT DATA			
A	m	$x_{1}$	VIG dimension in X-direction
B	m	$x_{2}$	VIG dimension in Y-direction
$t_{g}$	mm	$x_{3}$	Thickness of a glass pane
$n_{x}$	-	$x_{4}$	Number of pillars in X-direction
$n_{y}$	-	$x_{5}$	Number of pillars in Y-direction
$x_{p}$	mm	$x_{6}$	Pillars’ offset from the VIG edge in X-direction
$y_{p}$	mm	$x_{7}$	Pillars’ offset from the VIG edge in Y-direction
$f_{1}÷f_{30}$	Hz	$x_{8}÷x_{37}$	First thirty natural frequencies obtained from the numerical analysis
OUTPUT DATA			
*E_p_*	GPa	y	Young’s modulus of pillars’ material

**Table 5 materials-16-05055-t005:** Natural frequencies obtained from experimental analysis.

Plate No.	1	2	3	4	5	6	7
Mode No.	[Hz]	[Hz]	[Hz]	[Hz]	[Hz]	[Hz]	[Hz]
1	16	19	15	11	11	10	5
2	20	21	17	11	11	10	10
3	26	26	24	13	15	12	14
4	40	40	39	14	17	17	18
5	45	50	49	24	24	25	24
6	55	54	50	24	24	25	27
7	95	115	136	39	52	58	47
8	126	123	140	51	64	63	57
9	195	163	190	64	74	79	68
10	230	214	255	81	95	104	73
11	275	235	279	81	95	104	78
12	326	262	305	114	120	141	90
13	340	339	410	114	120	141	110
14	343	345	420	124	144	165	120
15	436	354	428	128	149	179	129
16	475	415	506	150	160	184	146
17	483	440	533	168	195	244	156
18	589	504	611	177	204	247	160
19	594	517	625	177	204	247	174
20	635	563	685	188	225	260	194
21	674	586	724	209	254	279	205
22	680	648	789	209	254	279	225
23	759	673	815	234	285	336	227
24	766	689	831	247	291	345	245
25	858	694	842	250	303	356	247
26	889	775	849	268	320	364	259
27	903	803	991	268	320	364	265
28	956	870	1070	288	342	402	282
29	1051	902	1100	298	349	412	292
30	1090	913	1111	317	366	435	295

**Table 6 materials-16-05055-t006:** VIG parameters combinations assumed for analysis.

Parameter	Unit	Assumed Values	Number of Combinations
A	m	0.30; 0.60; 0.90; 1.20; 1.50	15
B	m	0.30; 0.60; 0.90; 1.20; 1.50	
$t_{g}$	mm	4.0; 5.0; 6.0	3
$t_{v}$	mm	0.30	1
$w_{s}$	mm	9.0	1
$n_{x}$	-	Depended on *A* and *B*: 0.30–5; 0.60–10; 0.90–15; 1.20–21; 1.50–21	-
$n_{y}$	-	Depended on *A* and *B*: 0.30–5; 0.60–10; 0.90–15; 1.20–21; 1.50–21	
$d_{p}$	mm	0.6	1
$x_{p}$	mm	Depended on *A* and *B*: 0.30–40.0; 0.60–52.5; 0.90–65.0; 1.20–50.0; 1.50–62.5	-
$y_{p}$	mm	Depended on *A* and *B*: 0.30–40.0; 0.60–52.5; 0.90–65.0; 1.20–50.0; 1.50–62.5	
$\rho_{g}$	kg/m^3^	2500.0	1
$E_{g}$	GPa	72.0	1
$\nu_{g}$	-	0.22	1
$\rho_{p}$	kg/m^3^	7850.0	1
$\nu_{p}$	-	0.31	1
$\rho_{s}$	kg/m^3^	7850.0	1
$E_{s}$	GPa	210.0	1
$\nu_{s}$	-	0.31	1
*E_p_*	GPa	160.0; 170.0; 180.0; 190.0; 200.0; 210.0; 220.0	7
Total number of combinations			315

**Table 7 materials-16-05055-t007:** Assumed XGB parameters.

Parameter	Range of Values	Final Value
Number of estimators (decision trees)	500, 1000, 1500, 2000, 2500	2500
Gamma factor	0.0, 0.1, 0.2	0.0
Learning rate	0.01, 0.1, 0.2, 0.23, 0.26, 0.29, 0.32, 0.35	0.2
Maximum depth of a single tree	3, 5, 10, 20, 30	10
Grow policy	Depthwise, Lossguide	Depthwise
Colsample by tree	0.3, 0.6, 1.0	0.6
Subsample	0.3, 0.6, 1.0	0.6

**Table 8 materials-16-05055-t008:** The errors for XGB predictive model.

Data Set	RMSE [GPa]
Train data set	0.0005
Test data set	7.6107

**Table 9 materials-16-05055-t009:** Assumed DNN parameters.

Parameter	Range of Values	Final Value
Number of hidden layers	5, 10, 20, 30, 50	5
Size of a hidden layer	5, 10, 25, 50, 100	100
Learning rate	0.1, 0.05, 0.01, 0.005, 0.001	0.1
Activation function	ReLU, Linear, SELU	SELU
Droput Probability	0.1, 0.2, 0.3	0.0
Batchnorm usage	True, False	True
Weight decay	0.1, 0.01, 0.001, 0.0001	0.01
Batch size	32, 64, 128	128
Number of epochs	250, 500, 750, 1000	500

**Table 10 materials-16-05055-t010:** The errors for DNN predictive model.

Data Set	RMSE [GPa]
Train data set	20.4017
Test data set	18.6143

**Table 11 materials-16-05055-t011:** ML models predictions for the tested VIG plates.

No.	XGB Predictive Model [GPa]	DNN Predictive Model [GPa]	Expected Value [GPa]	XGB Relative Error [%]	DNN Relative Error [%]
1	208.65	193.52	200.00	4.33	3.35
2	196.55	186.55	200.00	1.76	7.21
3	212.32	192.91	200.00	6.16	3.68
4	188.02	189.54	200.00	6.37	5.52
5	182.54	192.88	200.00	9.57	3.69
6	171.83	189.73	200.00	16.39	5.41
7	185.71	189.80	200.00	7.69	5.37

## Data Availability

The data presented in this study are available on request from the corresponding author.

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
