# Peer review of "The Machine Learning Methods in Non-Destructive Testing of Dynamic Properties of Vacuum Insulated Glazing Type Composite Panels"

_materials, 2023, doi:10.3390/ma16145055_

Round 1

Reviewer 1 Report

This article presents a study where Machine Learning (ML) methods are employed to predict the elastic modulus of Vacuum Insulated Glazing (VIG). The prediction models are based on the geometric parameters of the structure and the first thirty natural frequencies obtained from numerical analysis. Deep Neural Networks and Extreme Gradient Boosting techniques are utilized, alongside the Finite Element Method and experimental data. The results obtained from these approaches are both scientifically and practically valuable. However, significant revisions are recommended before recommending the article for publication.

1. The reviewer recommends to use the term "Machine Learning" (ML) instead of "Artificial Intelligence" (AI), as well as to change the title of the article accordingly.

2. Section 2.1 combines Figures 1 and 2 into a single figure, which is placed in this section. Similarly, Figures 3, 4, and 5 are combined into one, while Figures 6 and 7 are also merged into a single figure. Additionally, Figures 14-21 have been improved by reducing the line thickness for better clarity.

3. The reviewer suggests transforming the article from a detailed study report into a format that emphasizes the novel scientific results. Excessive tables, known formulas, and other standard explanations should be moved to the accompanying material.

4. The article now doesn't include a clear and concise formulation of the problem concerning the use of ML methods. It doesn't discusse the significance of initial parameters and whether the elastic modulus predictions could have been made using classical regression models. The necessity of employing two methods isn't explained, including the array size used for training and testing. Furthermore, the prediction results from both models and their comparison with experimental data aren't presented explicitly. Table 11 shows only one expected value (200 GPa) for all seven cases analyzed?

5. English language needs to be significantly improved.

Authors should contact a native speaker for help.

Author Response

Response to the Reviewers concerning the manuscript entitled
The artificial intelligence methods in non-destructive testing of
dynamic properties of VIG type composite panels

by Damian Kozanecki, Izabela Kowalczyk, Sylwia Krasoń, Martyna Rabenda, Łukasz Domagalski, Artur Wirowski

05th of July, 2023

We would like to thank the Reviewers for their thorough review of our manuscript. They raised important issues and their contribution is very valuable for improvement in the quality of our manuscript. We generally agree with the Reviewers’ comments and we have revised our manuscript accordingly. We did our best to improve the manuscript greatly such that it meets the expectations of the Reviewers. Below we reply closely to every received comment. We hope that the Editor and the Reviewers will acknowledge our responses to their comments as satisfactory. We are ready to finish the revised version of the manuscript concerning any further suggestion that the Editor or Reviewers may have.

Please find our responses to your comments below.

We are looking forward to hearing from you soon.

Yours sincerely,

Damian Kozanecki, Izabela Kowalczyk, Sylwia Krasoń,

Martyna Rabenda, Łukasz Domagalski, Artur Wirowski

Reviewer #1:

This article presents a study where Machine Learning (ML) methods are employed to predict the elastic modulus of Vacuum Insulated Glazing (VIG). The prediction models are based on the geometric parameters of the structure and the first thirty natural frequencies obtained from numerical analysis. Deep Neural Networks and Extreme Gradient Boosting techniques are utilized, alongside the Finite Element Method and experimental data. The results obtained from these approaches are both scientifically and practically valuable. However, significant revisions are recommended before recommending the article for publication.

1. The reviewer recommends to use the term "Machine Learning" (ML) instead of "Artificial Intelligence" (AI), as well as to change the title of the article accordingly.

It has been changed.

2. Section 2.1 combines Figures 1 and 2 into a single figure, which is placed in this section. Similarly, Figures 3, 4, and 5 are combined into one, while Figures 6 and 7 are also merged into a single figure. Additionally rnie, Figures 14-21 have been improved by reducing the line thickness for better clarity.
Figures 14-21 have been improved.

  1. The reviewer suggests transforming the article from a detailed study report into a format that emphasizes the novel scientific results. Excessive tables, known formulas, and other standard explanations should be moved to the accompanying material.

Thank you for your suggestion. We have shortened some of the descriptions. Nonetheless, we reckon tables and formulas make this article more precise.

  1. The article now doesn't include a clear and concise formulation of the problem concerning the use of ML methods. It doesn't discusse the significance of initial parameters and whether the elastic modulus predictions could have been made using classical regression models. The necessity of employing two methods isn't explained, including the array size used for training and testing. Furthermore, the prediction results from both models and their comparison with experimental data aren't presented explicitly. Table 11 shows only one expected value (200 GPa) for all seven cases analyzed?
    The significance of initial parameters has been described in section 2.4.3. The reason for using two different methods has been added to section 2.4.3.
    Table 11 shows the comparison between the value predicted by XGB model, and DNN model, and.expected (anticipated – according to the technical data sheet) value of the pillars Young’s Modulus.

  2. English language needs to be significantly improved.

The whole manuscript was double-checked in order to eliminate all grammar mistakes. All the changes are marked with color in the main manuscript.

Reviewer 2 Report

The reviewed paper concerns with analyzing the VIG (Vacuum Insulated Glazing), that is actually the composite glazing, in which the space between glass panes is filled with vacuum,. The key elements of the VIG plates are the support pillars. The authors claim that, the most important and most probable issue lies in the analysis of their mechanical properties, (e.g. Young's modulus and it's variability over a long period of time). Artificial intelligence (AI) methods are used for numerically analysis of this problem. Among the many different techniques included in AI, neural networks (NN) and the extreme gradient boosting (XGB) algorithms are used. 

While the reviewed paper can be interesting for a potential reader of the journal, a revision is still needed.

1. The numerical results obtained by FE algorithms, are prone to the mesh size, in this respect it is advisable to give at least a comment on mesh convergence.

2. Claiming that computations were performed in a multigrid environment, it is advisable to give the timing dependence on number of computational cores etc. 

Author Response

Response to the Reviewers concerning the manuscript entitled
The artificial intelligence methods in non-destructive testing of
dynamic properties of VIG type composite panels

by Damian Kozanecki, Izabela Kowalczyk, Sylwia Krasoń, Martyna Rabenda, Łukasz Domagalski, Artur Wirowski

05th of July, 2023

We would like to thank the Reviewers for their thorough review of our manuscript. They raised important issues and their contribution is very valuable for improvement in the quality of our manuscript. We generally agree with the Reviewers’ comments and we have revised our manuscript accordingly. We did our best to improve the manuscript greatly such that it meets the expectations of the Reviewers. Below we reply closely to every received comment. We hope that the Editor and the Reviewers will acknowledge our responses to their comments as satisfactory. We are ready to finish the revised version of the manuscript concerning any further suggestion that the Editor or Reviewers may have.

Please find our responses to your comments below.

We are looking forward to hearing from you soon.

Yours sincerely,

Damian Kozanecki, Izabela Kowalczyk, Sylwia Krasoń,

Martyna Rabenda, Łukasz Domagalski, Artur Wirowski

Reviewer #2:

The reviewed paper concerns with analyzing the VIG (Vacuum Insulated Glazing), that is actually the composite glazing, in which the space between glass panes is filled with vacuum,. The key elements of the VIG plates are the support pillars. The authors claim that, the most important and most probable issue lies in the analysis of their mechanical properties, (e.g. Young's modulus and it's variability over a long period of time). Artificial intelligence (AI) methods are used for numerically analysis of this problem. Among the many different techniques included in AI, neural networks (NN) and the extreme gradient boosting (XGB) algorithms are used. 

While the reviewed paper can be interesting for a potential reader of the journal, a revision is still needed.

  1. The numerical results obtained by FE algorithms, are prone to the mesh size, in this respect it is advisable to give at least a comment on mesh convergence.

The description has been added in 3.2 section. The analysis is described in details in the section 2.2 of the paper Kowalczyk I, Kozanecki D; Krasoń S, Rabenda M. Computational Modelling of VIG Plates Using FEM: Static and Dynamic Analysis. Materials 2022, 15, 1467.

  1. Claiming that computations were performed in a multigrid environment, it is advisable to give the timing dependence on number of computational cores etc. 

The description of the duration of the calculations could be found in Section 2.3.

Reviewer 3 Report

The authors consider the practically important problem of assessing the state of VIG double-glazed windows by methods of non-destructive testing using artificial intelligence. The correct approach was chosen, combining experimental studies of the frequency spectrum of the response of the structure to dynamic impact, numerical FEM modeling and machine learning methods. Each stage of the study according to the parameters of the methods used is quite complete. The results obtained will be of interest to specialists in the field of applying artificial intelligence methods to the problems of assessing the condition of building structures.

However, there are a number of significant comments on the content of the article:

1.     From the review of literature sources, it is unclear why the elastic modules of the support pillars are the elements of forecasting. Over time, the mechanical properties of the glass may also change (there is an accumulation of microdefects), as well as the contact conditions of various structural elements and its depressurization. These changes may have a greater impact on the spectral composition of the VIG response of the double-glazed window. It is necessary to give a more complete analysis (description) of the mechanisms and causes of changes in the strength of VIG double-glazed windows, as well as an assessment at the level of FEM modeling of the sensitivity of natural oscillation frequencies to changes in these parameters.

2.     The article does not provide a justification for the number of oscillation forms used (30). The maximum natural oscillation frequency should be correlated primarily with the dimensions of the investigated structural element. It is also required to analyze the practical accuracy of the result obtained from the number of oscillation forms taken into account.

3.     Some parts of the article contain well-known facts, for example, a description of neural networks (lines 346-394), and can be somewhat shortened.

4.     The mathematical description of methods and models contains a significant number of inaccuracies:

a)     Formula (2): in the left part, most likely, should contain Xk (x large)

b)    Formula (5): in the right part, the exponent argument is written at the wrong level.

c)     Formula (18): the second term on the right side should have a multiplier  ft (xiinstead of  fi (xi )   

d)  In line (381), the shift coefficient w0  should be included in the expression for the vector W,   x0   or 1 in the expression for X in line (379).

e)     The presentation of the method of error back propagation chosen by the authors, starting with formula (29), is extremely unsuccessful. The division of neurons into sets (line 407) leads to duplication of the designations of elements belonging to different sets. This distorts the understanding of the dependencies between the input and output values of neurons. It is more correct to talk not about the number of the learning step, but about the number of the layer of neurons of a fully connected network. In this case, formula (29) looks like  ym(j)=xi(j+1)

 without dividing the neurons into any sets. In this case, the index j also corresponds to the step of the error back propagation process when it is changed in reverse order. The remaining formulas (30)-(35) also become correct. And the relations (31), (34) can not be duplicated. Formula (35) is generally written out incorrectly.

5.     The drawings of frequency spectra (14-21) are not informative enough. The graphs merge. Perhaps it is necessary to give frequency dependencies not on the entire interval of their change, but in the vicinity of 1 and 2 natural frequencies. At the same time, the author's conclusion about the need to take into account this behavior when using machine learning methods is absolutely correct. Frequency intervals (0,w1) or (w1w2) are most often used.  However this requires a separate study in this problem.

Conclusion:

The article can be accepted for publication in the Materials journal after major revision.

Author Response

Response to the Reviewers concerning the manuscript entitled
The artificial intelligence methods in non-destructive testing of
dynamic properties of VIG type composite panels

by Damian Kozanecki, Izabela Kowalczyk, Sylwia Krasoń, Martyna Rabenda, Łukasz Domagalski, Artur Wirowski

05th of July, 2023

We would like to thank the Reviewers for their thorough review of our manuscript. They raised important issues and their contribution is very valuable for improvement in the quality of our manuscript. We generally agree with the Reviewers’ comments and we have revised our manuscript accordingly. We did our best to improve the manuscript greatly such that it meets the expectations of the Reviewers. Below we reply closely to every received comment. We hope that the Editor and the Reviewers will acknowledge our responses to their comments as satisfactory. We are ready to finish the revised version of the manuscript concerning any further suggestion that the Editor or Reviewers may have.

Please find our responses to your comments below.

We are looking forward to hearing from you soon.

Yours sincerely,

Damian Kozanecki, Izabela Kowalczyk, Sylwia Krasoń,

Martyna Rabenda, Łukasz Domagalski, Artur Wirowski

Reviewer #3:

The authors consider the practically important problem of assessing the state of VIG double-glazed windows by methods of non-destructive testing using artificial intelligence. The correct approach was chosen, combining experimental studies of the frequency spectrum of the response of the structure to dynamic impact, numerical FEM modeling and machine learning methods. Each stage of the study according to the parameters of the methods used is quite complete. The results obtained will be of interest to specialists in the field of applying artificial intelligence methods to the problems of assessing the condition of building structures.

However, there are a number of significant comments on the content of the article:

  1. From the review of literature sources, it is unclear why the elastic modules of the support pillars are the elements of forecasting. Over time, the mechanical properties of the glass may also change (there is an accumulation of microdefects), as well as the contact conditions of various structural elements and its depressurization. These changes may have a greater impact on the spectral composition of the VIG response of the double-glazed window. It is necessary to give a more complete analysis (description) of the mechanisms and causes of changes in the strength of VIG double-glazed windows, as well as an assessment at the level of FEM modeling of the sensitivity of natural oscillation frequencies to changes in these parameters.

We agree with Reviewer’s opinion. However the most difficult task is to do research of pillars parameters and that is why we focused on this element. Defects of other parts of VIG plate may be investigated with other methods.

  1. The article does not provide a justification for the number of oscillation forms used (30). The maximum natural oscillation frequency should be correlated primarily with the dimensions of the investigated structural element. It is also required to analyze the practical accuracy of the result obtained from the number of oscillation forms taken into account.

We have chosen this number of oscillation forms in order to see behaviour of the element in wider spectrum. However, we agree with the Reviewer’s statement that 1st and 2nd natural frequencies have the greatest importance and accuracy.

  1. Some parts of the article contain well-known facts, for example, a description of neural networks (lines 346-394), and can be somewhat shortened.

Description in section 2.4.1 and 2.4.2 have been shortened.

  1. The mathematical description of methods and models contains a significant number of inaccuracies:   
  2. a) Formula (2): in the left part, most likely, should contain Xk (x large)

It has ben corrected.

b) Formula (5): in the right part, the exponent argument is written at the wrong level.

It has ben corrected.

c) Formula (18): the second term on the right side should have a multiplier ft (xi )  instead of  fi (xi )  

It has been corrected.

d) In line (381), the shift coefficient w0 should be included in the expression for the vector W,   x0   or 1 in the expression for X in line (379).

It has ben corrected.

e) The presentation of the method of error back propagation chosen by the authors, starting with formula (29), is extremely unsuccessful. The division of neurons into sets (line 407) leads to duplication of the designations of elements belonging to different sets. This distorts the understanding of the dependencies between the input and output values of neurons. It is more correct to talk not about the number of the learning step, but about the number of the layer of neurons of a fully connected network. In this case, formula (29) looks like ym(j)=xi(j+1) without dividing the neurons into any sets. In this case, the index j also corresponds to the step of the error back propagation process when it is changed in reverse order. The remaining formulas (30)-(35) also become correct. And the relations (31), (34) can not be duplicated. Formula (35) is generally written out incorrectly.

The description of the method is based on the Polish book entitled “Sieci Neuronowe” - Sieci Neuronowe. Akademicka Oficyna Wydawnicza, 1993.

  1. The drawings of frequency spectra (14-21) are not informative enough. The graphs merge. Perhaps it is necessary to give frequency dependencies not on the entire interval of their change, but in the vicinity of 1 and 2 natural frequencies. At the same time, the author's conclusion about the need to take into account this behavior when using machine learning methods is absolutely correct. Frequency intervals (0,w1) or (w1, w2) are most often used. However this requires a separate study in this problem.

The mentioned figures were revised in order to clarify the presented data.

The drawings 14-21 (in corrected version 13-20) show the data from the LAN-XI type 3160 measuring cassette acquisition system. We definitely agree that frequency intervals (0, w1) or (w1, w2) are most often used. In order to to that we would need to reconstruct the study. However, the laboratory research was done in 2021 and there is no possibility to do further research on that as test stand  no longer exists.

Reviewer 4 Report

The paper is well written, however, the following revisions are required before acceptance.

1) In the introduction, first please start with a paragraph and mention what you are going to perform, then itemize the topics.

2) Preferably please revisit the structure of the introduction.

3) Please add descriptions to the figures and add details to them. Also, they are very big, and they are not in good shape.

4) Figure 8 does not make sense to show a screenshot of a software window.

5) The number of cited articles is very small. You may add articles about using ABAQUS and artificial intelligence. For ABAQUS,  you may use these papers if you see fit.

https://www.nature.com/articles/s41598-022-14685-x

https://link.springer.com/article/10.1007/s00366-020-00954-8

https://moodle.umontpellier.fr/pluginfile.php/480056/mod_resource/content/0/Sun-ShearLocking-Hourglassing.pdf

6) Conclusion is not concise, please try to give a summary of work only with concluding remarks.

Author Response

Response to the Reviewers concerning the manuscript entitled
The artificial intelligence methods in non-destructive testing of
dynamic properties of VIG type composite panels

by Damian Kozanecki, Izabela Kowalczyk, Sylwia Krasoń, Martyna Rabenda, Łukasz Domagalski, Artur Wirowski

05th of July, 2023

We would like to thank the Reviewers for their thorough review of our manuscript. They raised important issues and their contribution is very valuable for improvement in the quality of our manuscript. We generally agree with the Reviewers’ comments and we have revised our manuscript accordingly. We did our best to improve the manuscript greatly such that it meets the expectations of the Reviewers. Below we reply closely to every received comment. We hope that the Editor and the Reviewers will acknowledge our responses to their comments as satisfactory. We are ready to finish the revised version of the manuscript concerning any further suggestion that the Editor or Reviewers may have.

Please find our responses to your comments below.

We are looking forward to hearing from you soon.

Yours sincerely,

Damian Kozanecki, Izabela Kowalczyk, Sylwia Krasoń,

Martyna Rabenda, Łukasz Domagalski, Artur Wirowski

Reviewer #4:

The paper is well written, however, the following revisions are required before acceptance.

  1. In the introduction, first please start with a paragraph and mention what you are going to perform, then itemize the topics.

The order of the introduction section has been changed.

  1. Preferably please revisit the structure of the introduction.

The structure of the introduction has been revised.

  1. Please add descriptions to the figures and add details to them. Also, they are very big, and they are not in good shape.

The description of several figures has been detailed and the size of. the figures has been changed

  1. Figure 8 does not make sense to show a screenshot of a software window.

The mentioned figure has been deleted.

  1. The number of cited articles is very small. You may add articles about using ABAQUS and artificial intelligence. For ABAQUS, you may use these papers if you see fit.

https://www.nature.com/articles/s41598-022-14685-x

https://link.springer.com/article/10.1007/s00366-020-00954-8

 https://moodle.umontpellier.fr/pluginfile.php/480056/mod_resource/content/0/Sun-ShearLocking-Hourglassing.pdf

We see the paper https://moodle.umontpellier.fr/pluginfile.php/480056/mod_resource/content/0/Sun-ShearLocking-Hourglassing.pdf fit do our article and we added it in the section 2.3

  1. Conclusion is not concise, please try to give a summary of work only with concluding remarks.

Short summary with the concluding remarks have been added.

Round 2

Reviewer 1 Report

I have read the revised version and I think the article can be published as it is.

I have read the revised version and I think the article can be published as it is.

Reviewer 3 Report

The content of the article has been improved in accordance with the comments. Some issues require further study. I hope for the continuation of the authors' research on the chosen topic.

Reviewer 4 Report

Paper is accepted in this format.